# Transient neurological symptoms in the older population: report of a prospective cohort study—the Medical Research Council Cognitive Function and Ageing Study (CFAS)

Nahal Mavaddat,[1] George M Savva,[2] Daniel S Lasserson,[3] Matthew F Giles,[4] Carol Brayne,[5] Jonathan Mant[1]

For numbered affiliations see end of article.

Correspondence to
Dr Nahal Mavaddat;
nm212@medschl.cam.ac.uk

## ABSTRACT

**Objective:** Transient ischaemic attack (TIA) is a recognised risk factor for stroke in the older population requiring timely assessment and treatment by a specialist. The need for such TIA services is driven by the epidemiology of transient neurological symptoms, which may not be caused by TIA. We report prevalence and incidence of transient neurological symptoms in a large UK cohort study of older people.

**Design:** Longitudinal cohort study

**Setting:** The Medical Research Council Cognitive Function and Aging Study (CFAS) is a population representative study based on six centres across England and Wales.

**Participants:** Random samples of people in their 65th year were obtained from Family Health Service Authority lists. The participation rate was 80% (n=13 004). Interview at baseline included questions about stroke and three transient neurological symptoms, repeated in a subsample after 2 years. Patients were flagged for mortality.

**Main outcome measures:** Prevalence and 2-year incidence of transient neurological symptoms.

**Results:** In 11 903 participants without a history of stroke, 271 (2.3%) reported transient problems with speech, 872 (7.6%) with sight and 596 (5.1%) weakness in a limb with 1456 (12.7%) reporting at least one symptom. Of those reinterviewed (n=6748), 675 (9.8%) reported at least one symptom over 2 years.

**Conclusions:** Lifetime prevalence and incidence of transient neurological symptoms in people aged 65 years and over is high and is substantially greater than the incidence of TIA in hospital-based and population-based studies. These high rates of transient neurological symptoms in the community in the older population should be considered when planning TIA services.

## ARTICLE SUMMARY

### Article focus
- Prompt initiation of secondary prevention following transient ischaemic attack (TIA) is associated with up to 80% reduction in risk of subsequent stroke.
- Many people presenting to specialist TIA clinics with transient neurological symptoms do not have TIA.
- The prevalence and incidence of transient neurological symptoms (as opposed to TIA) in older age groups is unknown.

### Key messages
- In a large multicentred community-based study representative of the older population in the UK, we found a high prevalence and incidence of transient neurological symptoms, significantly greater than that of TIAs in hospital-based and population-based studies.
- These high rates of transient neurological symptoms in the community in the older population should be considered when planning TIA services.

### Strengths and limitations of this study
- Previous studies of incidence and prevalence of transient neurological symptoms have mostly been conducted in populations not representative of older age groups where TIA is most common and few have been undertaken in UK populations.
- The incidence of transient neurological symptoms in the study was determined in people without severe cognitive impairment, and may be an underestimation, since cognitive impairment can be a manifestation of vascular disease. The wording of questions to identify transient neurological symptoms may have picked up some people who had symptoms lasting more than 24 h.

## INTRODUCTION

Transient ischaemic attack (TIA) is an established and powerful risk factor for stroke. Eight per cent of patients who have TIA suffer from stroke within 7 days, many within 48 h.[1][2] Immediate specialist assessment and treatment is associated with substantial reductions in this early risk of stroke,[3] and is recommended for people with suspected TIA.[4]

As awareness grows of the urgency of early management of TIA among both primary care practitioners and the general public,[5] more patients with symptoms suggestive of TIA are likely to present to primary and secondary care services. Need for specialist services will be driven by incidence of symptoms that might represent TIA (ie, transient neurological symptoms) rather than the epidemiology of TIA per se. Currently about 48 000 probable or definite TIAs (transient neurological symptoms lasting less than 24 h of likely vascular aetiology) and 43 000 minor strokes (stroke events causing minimal or no neurological deficits) are managed as outpatients every year in England, but it is not clear to what extent this might be the tip of the iceberg.[6] Data are available worldwide on the prevalence and incidence of TIA.[7–11] The epidemiology in the community of transient neurological symptoms (ie, neurological symptoms of sudden onset of which TIA is a subset) is however less well defined. Previous studies of incidence and prevalence of transient neurological symptoms have mostly been conducted in populations not representative of older age groups where TIA is most common.[7 11–16] Very few have been undertaken in UK populations. While incidence of transient neurological symptoms will drive the need for specialist TIA services, prevalence of transient neurological symptoms in the community is also important. TIA is associated with a long-term increase in risk of stroke,[17][18] so there may be value in diagnosing 'old' events that have not presented to medical services in order to target secondary prevention. Hence, the importance of determining both the incidence and prevalence of transient neurological symptoms in the community.

This study reports prevalence and incidence of three common transient neurological symptoms (limb weakness, loss of speech and disturbance of vision) in a population-based multicentred cohort study in the UK (England and Wales) in those aged 65 years and over, the Medical Research Council Cognitive Function and Ageing Study (CFAS).[19]

## METHODS

The Medical Research Council CFAS is a population-representative study of individuals aged 65 years and over. The study began in 1991 and was designed to determine the incidence of dementia in the older population. It has six centres across England and Wales chosen to represent the national variation of urban–rural mix, socioeconomic deprivation and rates of chronic disease.[19] Five of these with identical study designs (Oxford, Nottingham, Cambridgeshire, Gwynedd and Newcastle) are used in the present investigation. The sixth centre (Liverpool) used a different design and is therefore excluded. Random samples of people in their 65th year and above were obtained from Family Health Service Authority lists (agency responsible for maintaining registers of general practice populations at that time). The sample was stratified by age (65–74, 75 years and over) and equal numbers were randomly selected from these groups with the aim of recruiting 2500 to each centre. All study centres obtained ethical approval from local research committees (REC Ref: 05/MRE05/37). Full details of methodology are available elsewhere.[19]

Eligible participants (or their proxies where appropriate) provided informed consent. Trained interviewers undertook baseline interviews in the participants' homes, including sociodemographic characteristics, cognitive function and disease history, including previous stroke, coronary heart disease and diabetes (full details at http://www.cfas.ac.uk). On the basis of baseline screening, the study sample was divided into two groups at baseline: people without cognitive impairment and a group consisting of those with cognitive impairment plus a stratified subsample of those without cognitive impairment. The first group underwent no further assessment at baseline and was 'rescreened' after 2 years. The second group underwent a further detailed cognitive assessment at baseline, but was not followed up for transient neurological symptoms. The estimates of prevalence of transient neurological symptoms are calculated for all people at baseline, while the incidence estimates use only the first group. All participants were flagged with the National Health Service (NHS) Central Register. Deaths and underlying causes of death attributed to stroke (International Classification of Diseases (ICD) codes 430–438) were notified to the study.

### Prevalence of transient neurological symptoms

At baseline all participants were asked "Have you ever experienced sudden problems with:
1. speech, which got better after a day?
2. weakness in the arms or legs, which got better after a day?
3. sight, which got better after a day?"

Social class was determined using the Registrar General's Occupational Classification. Cognitive status was determined using the Mini-Mental State Examination (MMSE)[20] and the Automated Geriatric Examination for Computer Assisted Taxonomy (AGECAT).[21]

### Incidence of transient neurological symptoms

All those in the 'rescreen' arm were asked at 2-year follow-up if they had experienced each transient neurological symptom in the past 2 years.

### ANALYSIS

All analyses were carried out using STATA V.11. Inverse probability weights were used throughout to ensure that

the sample was representative of the target population. Weights were estimated with logistic regression using presence in each phase of the study as an outcome and taking into account oversampling of over-75s at baseline. Weights for incidence calculations adjust for attrition based on baseline characteristics and stratified selection into the assessment arm as appropriate. Baseline prevalence of each transient neurological symptom was calculated for age-specific strata and as a weighted percentage to provide a population estimate of prevalence for people aged 65 years and over. Associations with gender, age, social class (manual (IIIb, IV and V) and non-manual (I, II and IIIa)) and cardiovascular comorbidity were explored using logistic regression models adjusting for all other factors and cognitive function based on MMSE by score (less than 18, 18–21, 22–25 and 26–30).

Two-year incidence of transient neurological symptoms was estimated using weighted percentages of those reporting any of the symptoms during follow-up. Weights were adjusted for refusals, dropouts and for non-reassessment of people with cognitive impairment. Calculation of attrition weights for incident transient neurological symptoms excluded those who died between baseline and follow-up. These estimates are therefore applicable to the population over 65 years without severe cognitive impairment or stroke at baseline surviving at least 2 years.

## RESULTS
The participation rate in the CFAS was 80% (13 004/16 258). For the prevalence analysis, participants who had a stroke at baseline (963, 7.4%) or for whom baseline information about stroke was missing (138, 1.1%) were excluded, leaving 11 903 participants. In total, 2283 of the original participants were not allocated to the 'rescreen' arm at 2 years; 754 died, 1973 declined to participate and 145 were lost to follow-up, leaving 6748 (76% response rate of potential participants) for the incidence analysis.

Table 1 shows the demographic features of participants in the CFAS, studied in this analysis at baseline. Forty per cent of participants were men and 60% were women. 12.3% of all participants reported at least one transient neurological symptom; 2.3% reported transient loss of speech, 5.1% transient weakness and 7.6% transient loss of sight.

Table 2 shows prevalence data with weighted percentages and ORs (adjusted for age, sex, social class, cognition and cardiovascular morbidities) of reported transient neurological symptoms at baseline by demographic factors and comorbidities. In total, 12.8% of men and 12% of women reported at least one of the three transient neurological symptoms of loss of speech, loss of sight or weakness. There was no significant association of gender with reporting at least one transient neurological symptom, but significantly lower odds of reporting a transient neurological symptom in the over

**Table 1** Demographics of participants in the Cognitive Function and Aging Study (CFAS) by sex, age, social class and comorbidity (data are counted in percentages)

| | All (n=11903) | | Males (n=4689) | | Females (n=7214) | |
|---|---|---|---|---|---|---|
| | N | Per cent* | N | Per cent* | N | Per cent* |
| Age (years) | | | | | | |
| 65–74 | 5980 | 58.8 | 2635 | 64.5 | 3345 | 55.0 |
| 75–84 | 4624 | 32.1 | 1716 | 29.6 | 2908 | 33.8 |
| 85+ | 1299 | 9.1 | 338 | 5.9 | 961 | 11.2 |
| Social class | | | | | | |
| I | 558 | 4.8 | 225 | 4.8 | 333 | 4.8 |
| II | 2988 | 26.1 | 1213 | 26.3 | 1775 | 25.9 |
| IIIa | 1337 | 11.6 | 456 | 9.9 | 881 | 12.8 |
| IIIb | 4295 | 37.7 | 1865 | 40.8 | 2430 | 35.5 |
| IV | 1727 | 14.9 | 650 | 13.9 | 1077 | 15.7 |
| V | 561 | 4.9 | 196 | 4.3 | 365 | 5.3 |
| Comorbidity | | | | | | |
| Angina | 1923 | 16.3 | 878 | 15.3 | 1045 | 11.1 |
| Diabetes | 672 | 5.5 | 310 | 6.4 | 362 | 4.9 |
| Heart attack | 1139 | 9.5 | 632 | 13.6 | 507 | 6.8 |

*Weighted percentages.

85 age group. The odds of reporting any transient neurological symptom were higher in those in manual compared to non-manual social classes. The presence of cardiovascular morbidities of angina and heart attack were also significantly associated with the odds of having at least one of the transient neurological symptoms.

There was no difference between genders with regard to reporting individual transient neurological symptoms of loss of speech, loss of sight or weakness. However, there was a significantly lower odds of reporting transient visual loss in those aged over 85 years, and an increased odds of reporting transient symptoms of weakness and loss of sight in manual compared to non-manual social classes. The presence of cardiovascular comorbidities of angina and heart attack were associated with higher odds of all three symptoms.

Table 3 shows 2-year incidence of transient neurological symptoms in respondents attending the 'rescreen'. A total of 9.8% of participants reported at least one transient neurological symptom over the past 2 years, with the highest incidence of reported transient neurological symptoms being due to loss of sight followed by weakness, with loss of speech being the least frequently reported transient neurological symptom. The incidence of each of the transient neurological symptoms was highest in the 75–84 age group and lowest in the over 85 age group.

## DISCUSSION
Our findings suggest that transient neurological symptoms are common in the older population in England

**Table 2** Distribution and adjusted odds of transient neurological symptoms in those aged 65 years or over in the Cognitive Function and Aging Study (CFAS) by sex, social class and comorbidity (data are counted in percentages)

| | Loss of speech | | | Weakness | | | Loss of sight | | | At least one symptom | | |
|---|---|---|---|---|---|---|---|---|---|---|---|---|
| | N | Per cent* | OR† (95% CI) | N | Per cent* | OR† (95% CI) | N | Per cent* | OR† (95% CI) | N | Per cent* | OR† (95% CI) |
| **Gender** | | | | | | | | | | | | |
| Male | 119 | 2.5 | 1.0 | 257 | 5.5 | 1.0 | 347 | 7.6 | 1.0 | 598 | 12.8 | 1.0 |
| Female | 152 | 2.2 | 0.9 (0.7 to 1.2) | 339 | 4.8 | 0.9 (0.7 to 1.0) | 525 | 7.6 | 1.0 (0.9 to 1.2) | 858 | 12.0 | 1.0 (0.9 to 1.1) |
| **Age (years)** | | | | | | | | | | | | |
| 65–74 | 137 | 2.3 | 1.0 | 302 | 5.1 | 1.0 | 480 | 8.1 | 1.0 | 768 | 12.9 | 1.0 |
| 75–84 | 104 | 2.3 | 0.7 (0.7 to 1.2) | 229 | 5.0 | 0.9 (0.7 to 1.1) | 327 | 7.2 | 0.9 (0.7 to 1.1) | 553 | 12.0 | 0.9 (0.8 to 1.0) |
| 85+ | 30 | 2.5 | 0.8 (0.5 to 1.3) | 65 | 5.4 | 0.9 (0.6 to 1.2) | 65 | 5.4 | 0.6 (0.5 to 0.8) | 134 | 10.4 | 0.7 (0.6 to 0.9) |
| **Social class‡** | | | | | | | | | | | | |
| Non-manual | 96 | 5.8 | 1.0 | 63 | 11.0 | 1.0 | 295 | 17.9 | 1.0 | 497 | 10.2 | 1.0 |
| Manual | 164 | 7.2 | 1.1 (0.9 to 1.5) | 390 | 17.8 | 1.4 (1.2 to 1.7) | 556 | 25.7 | 1.4 (1.2 to 1.6) | 920 | 14.1 | 1.4 (1.2 to 1.6) |
| **Comorbidity** | | | | | | | | | | | | |
| Angina (n=1538) | 64 | 4.3 | 1.5 (1.0 to 2.2) | 176 | 11.7 | 2.5 (2.0 to 3.2) | 160 | 10.6 | 1.3 (1.0 to 1.6) | 312 | 20.7 | 1.7 (1.4 to 1.9) |
| Diabetes (n=672) | 28 | 4.0 | 1.4 (0.9 to 2.2) | 40 | 6.1 | 1.0 (0.7 to 1.4) | 66 | 10.1 | 1.3 (0.9 to 1.7) | 102 | 15.2 | 1.1 (0.9 to 1.4) |
| Heart attack (n=1139) | 52 | 4.7 | 1.8 (1.2 to 2.7) | 131 | 10.5 | 1.4 (1.2 to 1.9) | 119 | 11.5 | 1.4 (1.1 to 1.8) | 233 | 12.4 | 1.4 (1.2 to 1.8) |

*Weighted percentages.
†Adjusted for age, sex, social class, cognition and cardiovascular comorbidities.
‡Reference non-manual (I, II and IIIa) compared to manual social class (IIIb, IV and V).
156 participants had missing data on the speech questionnaire, 155 on the weakness and 154 on the sight question.

**Table 3** Two-year incidence of transient neurological symptoms in the Cognitive Function and Aging Study (CFAS) in those without stroke at baseline at 2-year screening assessment, N at 2 years was 6748 (data are in N (%))

| | Age (years) | | | | | | | |
| | All | | 65–74 (N=3657) | | 75–84 (N=2618) | | 85+ (N=473) | |
| Transient neurological symptoms | N | Per cent* | N | Per cent* | N | Per cent* | N | Per cent* |
|---|---|---|---|---|---|---|---|---|
| Loss of speech | 149 | 2.1 | 55 | 1.5 | 84 | 3.3 | 10 | 2.0 |
| Weakness | 239 | 3.7 | 117 | 3.4 | 106 | 4.1 | 16 | 3.4 |
| Loss of sight | 410 | 6.3 | 214 | 6.3 | 170 | 6.6 | 26 | 4.5 |
| At least one symptom | 675 | 9.8 | 327 | 9.3 | 304 | 11.6 | 44 | 7.8 |

*Weighted percentage.
98 participants had missing data on the speech questionnaire, 97 on the weakness and 105 on the sight question.

and Wales, with at least 12% of people aged 65 years and over having experienced a transient neurological symptom of the arm or leg, speech or vision, which gets better after a day, and approximately 5% having experienced at least one such symptom over the course of a year. The commonest of these was transient symptoms of vision, followed by limb weakness. Those aged over 85 years reported lower rates of transient neurological symptoms, predominantly due to less frequent reporting of transient visual loss. Problems of memory and recall may have contributed to under-reporting, and higher mortality rates associated with neurological incidents to underestimation of rates of transient neurological symptoms in this age group. Chronic visual problems may have also potentially masked transient visual losses in the oldest-old population. Incidence of at least one transient neurological symptom in the CFAS population was approximately 2.6 times greater than that of confirmed TIA presenting to medical services in the Oxford Vascular (OxVASC) study in people aged over 85 years (approximately 15/1000/year), eight times greater in those aged 75–84 years (7/1000/year) and approximately 15 times greater in those aged 65–74 years (3/1000 population/year).[10]

The prevalence and incidence of transient neurological symptoms in the CFAS was somewhat higher than those found by questionnaires used in other studies internationally (mostly conducted in younger age groups), but comparable to those of Wilkinson et al in an over 60 age group in a US-based study.[7 12–14 16] Online supplementary table S4 shows the prevalence of sudden onset of neurological symptoms of weakness in a limb or of loss of speech or sight in previous population and community studies.[7 12–14 16] Questions used to elicit transient neurological symptoms in the CFAS are likely to have captured greater numbers of transient neurological events in the population compared with those seeking more specific vascular symptoms, or to the few studies defining more precisely the onset, offset and timing of the event.[7 12–14] While some respondents will have been describing true transient ischaemic events, it is likely that many did not have a true TIA. Validated measures to determine the presence of previous TIA are limited, and most studies of transient neurological

symptoms in questionnaires overestimate true transient ischaemic attacks.[13 14 22] Wilkinson et al[14] suggest that around 10% of transient symptoms reported in a questionnaire are subsequently diagnosed as TIA after a neurologist's assessment. Self-reported transient neurological symptoms in other community-based studies have low positive predictive value, with higher values only in studies in outpatient populations (over 70%).[13 14 22–24] This suggests that the use of screening to identify possible past transient ischaemic attacks may generate unnecessary extra strain on primary care and secondary TIA services with limited benefit.

Patients presenting with transient neurological symptoms present considerable diagnostic dilemmas for primary care practitioners reflected in low rates of TIA diagnosis confirmation in patients referred to UK TIA clinics (over 50% being for non-TIA causes).[6 25 26] Many symptoms of TIA are non-specific and occur in non-vascular syndromes. Dizziness, ophthalmological problems, migraine, epilepsy, nerve entrapment or psychological states are the commonest non-TIA diagnoses.[25–29] Often, however, no diagnosis can be determined.[25 26] Scales such as the ABCD2[2] help determine the urgency of TIA referral, but do not distinguish TIA from non-TIA symptoms. Wilkinson et al[14] reported transient loss of speech and loss of sight to be more reliable than limb weakness for neurologist diagnosis of TIA, while Hart et al[7] found the strongest and most consistent relationship with subsequent stroke to be loss of power in an arm. Other studies suggest that the presence of certain symptoms such as headache, dizziness, loss of consciousness, memory loss, blurred vision, generalised weakness, pain in limbs and seizures make the diagnosis of TIA less likely.[14 29 30]

### Study limitations
The incidence of transient neurological symptoms was determined in people without severe cognitive impairment, and may be an underestimate since cognitive impairment can be a manifestation of vascular disease.[31 32] Only three transient neurological symptoms were reported in this study. Some symptoms of TIA (eg, posterior circulation or pure sensory symptoms) were not sought. The wording of questions to identify transient

neurological symptoms may have picked up some people who had symptoms lasting more than 24 h. Such patients are nevertheless likely to draw on TIA services for further assessment. In addition, it was not possible to determine whether different transient neurological symptoms in a participant occurred simultaneously or separately in time. The problem of recall bias in these self-reports of transient symptoms needs to be considered. Questions were designed to be easily comprehensible to the older population. However, the need to consider multiple criteria when responding to a question, for example, 'sudden onset' and 'less than a day', may have led to difficulties in interpretation of the transient neurological symptoms questions for some participants. Finally, baseline and 2-year follow-up were carried out in the 1990s and changes in the rates of vascular events may have occurred since then. While the age-specific incidence of stroke appears to be declining, it is not clear that the same is true of TIA.[10 33 34] This study is more recent than many studies estimating rates of transient neurological symptoms in the population, many of which have been carried out during the 1970s or 1980s.[12–14]

## CONCLUSION

In a large multicentred community-based study, highly representative of the older population in the UK and conducted in the age group where TIAs are most common, we found a high prevalence and incidence of transient neurological symptoms. The incidence of such symptoms in the community is significantly greater than that of TIAs. This highlights the need for adequate provision of TIA services and the potential importance of the development of valid diagnostic tools to assist the general practitioner and hospital doctor in better triage of people presenting with transient neurological symptoms.

**Author affiliations**
[1]Primary Care Unit, Department of Public Health and Primary Care, University of Cambridge, Strangeways Laboratory, Worts Causeway, Cambridge, UK
[2]School of Nursing Sciences, University of East Anglia, Norwich Research Park, Norwich, UK
[3]Department of Primary Care Health Sciences, University of Oxford, Oxford, UK
[4]Stroke Prevention Research Unit, Department of Clinical Neurology, John Radcliffe Hospital, NIHR Biomedical Research Centre, Oxford University, Oxford, UK
[5]Department of Public Health and Primary Care, University of Cambridge, Institute of Public Health, Cambridge, UK

**Acknowledgements** We would like to thank the MRC CFAS study group for data collection and management. We are also grateful to all respondents, their families and their primary care teams from across the country for their participation in the MRC CFAS.

**Contributors** NM and GMS contributed to analysis of data. All authors contributed to study design and intellectual input.

**Funding** The MRC CFAS has had funding by major awards from the Medical Research Council (MRC/G9901400) and Department of Health. NM was funded by a National Institute for Health Research (NIHR) Walport Clinical Lectureship in Primary Care Research.

**Competing interests** None.

**Patient consent** Obtained.

**Ethics approval** Multicentre REC (Ref: 05/MRE05/37).

**Provenance and peer review** Not commissioned; externally peer reviewed.

**Data sharing statement** Data can be shared through application. For further information please refer to the application form on the website http://www.cfas.ac.uk.

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
