## [Reviewer comments · BMJ Open]

Some articles will have been accepted based in part or entirely on reviews undertaken for other BMJ Group journals. These will be reproduced where possible.

ARTICLE DETAILS

TITLE (PROVISIONAL)	Transient neurological symptoms in the older population: Report of a prospective cohort study - the Medical Research Council Cognitive Function and Ageing (CFAS) Study
AUTHORS	Mavaddat, Nahal; Savva, George; Lasserson, Daniel; giles, matthew; Brayne, Carol; Mant, Jonathan

VERSION 1 - REVIEW

REVIEWER	Anthony S. Kim, MD, MAS Assistant Professor Department of Neurology University of California, San Francisco United States I have no competing interests.
REVIEW RETURNED	21-May-2013

THE STUDY	Supplemental documents are adequate, do not contain information that would be better reported in the manuscript, and do not raise questions about the work.
GENERAL COMMENTS	Nahal and colleagues have conducted an interesting population-based analysis of the incidence of transient neurological symptoms among older individuals in the UK using data collected from a large prospective cohort in the 1990s. The sampling methods are well-validated, the manuscript is clearly written, and the conclusions are well-tempered in the context of known limitations identified by the authors. A relatively high incidence of these transient neurological symptoms may have implications on future TIA referrals, though this speculation on this point goes beyond the scope of the current study. The wording of the instrument that was used to elicit the transient neurological symptoms may not be ideal: Each symptom question actually contains multiple criteria-- the symptoms must have been "sudden" AND have lasted for "less than a day" for example. These compound choices for each symptom increase the possibility of misinterpretation--particularly when a "yes" implies satisfying ALL criteria (including length and suddenness of onset of symptoms-- rather than just one of these criteria. However, since the instrument was administered by a trained interviewer, rather than by using simple questionnaire for self-report, could mitigate this issue. Any available information on the characteristics of these question during pre-testing of the instrument would be enlightening. One interesting analysis that is not included in the present study, would be to look for an association between these transient neurological symptoms and subsequent incident stroke or TIA, such as was done in Bos et al 2007 (JAMA. 2007;298(24):2877-2885.

	doi:10.1001/jama.298.24.2877.) Admittedly, the current data source may not be well-suited for this type of analysis though.
--	---

REVIEWER	Andrew Wilson Professor of Primary Care Research University of Leicester UK
REVIEW RETURNED	24-May-2013

GENERAL COMMENTS	CFAS provides a good opportunity to examine the incidence and prevalence of transient neurological symptoms (TNS) in UK older people, especially as there is a paucity of previous research in this area. The limitations of the study are due to the questions asked and the methodology of the surveys. However I do have the following suggestions to improve the clarity and interpretation of the paper.  1. It could be made more explicit, particularly in the abstract, that the study asked only about three TNS. Although this is mentioned in the 'limitations section', it is also relevant to the discussion when findings are compared to other studies that included a wider range of symptoms. 2. I think there should be less focus on the findings being directly relevant to supply of TIA services. For example visual symptoms do not feature in campaigns such as FAST and so may not present more often to medical services as a result of publicity. Furthermore, it is likely that many of the symptoms reported here would not be referred as possible TIA if they presented to primary care (for example there is no distinction between unilateral and bilateral). 3. Some of the text included in Results (page 10) is speculative and would be better placed in the discussion section. 4. In the first para of discussion, it would be worth emphasising that incidences in Oxvasc are cases presenting to medical services. 5. References 6 and 27 are the same.
--

VERSION 1 – AUTHOR RESPONSE

Dr. Kim

1. "A relatively high incidence of these transient neurological symptoms may have implications on future TIA referrals, though this speculation on this point goes beyond the scope of the current study."

We have revised the abstract and conclusions to reduce reference to predicted future increased need for TIA services.

2. The wording of the instrument that was used to elicit the transient neurological symptoms may not

be ideal: Each symptom question actually contains multiple criteria-- the symptoms must have been "sudden" AND have lasted for "less than a day" for example. These compound choices for each symptom increase the possibility of misinterpretation--particularly when a "yes" implies satisfying ALL criteria (including length and suddenness of onset of symptoms--rather than just one of these criteria. However, since the instrument was administered by a trained interviewer, rather than by using simple questionnaire for self-report, could mitigate this issue. Any available information on the characteristics of these question during pre-testing of the instrument would be enlightening.

Possible limitations in the wording of the questions which may result in misinterpretation as pointed out by Dr. Kim has been discussed in the limitations section of the discussion. Further information about characteristics of the questions during pre-testing was not available.

3. One interesting analysis that is not included in the present study, would be to look for an association between these transient neurological symptoms and subsequent incident stroke or TIA, such as was done in Bos et al 2007 (JAMA. 2007;298(24):2877-2885. doi:10.1001/jama.298.24.2877.)

We are able to perform the analysis suggested by Dr. Kim on the relationship between the transient neurological symptoms and subsequent incidence of stroke in the CFAS and hope to publish this data in a separate report.

In response to Professor Wilson's comments:

1. It could be made more explicit, particularly in the abstract, that the study asked only about three TNS. Although this is mentioned in the 'limitations section', it is also relevant to the discussion when findings are compared to other studies that included a wider range of symptoms.

It has been made more explicit in the abstract of the study and the discussion that only three transient neurological symptoms were elicited.

2. I think there should be less focus on the findings being directly relevant to supply of TIA services. For example visual symptoms do not feature in campaigns such as FAST and so may not present more often to medical services as a result of publicity. Furthermore, it is likely that many of the symptoms reported here would not be referred as possible TIA if they presented to primary care (for example there is no distinction between unilateral and bilateral).

The need for less focus on the findings being directly relevant to future TIA services has been addressed as above.

3. Some of the text included in Results (page 10) is speculative and would be better placed in the discussion section.

The speculative text in the results section on page 10 has been moved to the discussion section

4. In the first para of discussion, it would be worth emphasising that incidences in Oxvasc are cases presenting to medical services.

A phrase to emphasise that incidences in Oxvasc are cases presenting to medical services has been added to the discussion section in paragraph 1.

5. References 6 and 27 are the same.

Have corrected the repeated reference.

A few minor wording changes have also made to the text to improve clarity.